

# The extent to which cancer patients trust in cancer-related online information: a systematic review

Lukas Lange, Mona Leandra Peikert, Christiane Bleich and Holger Schulz

Department of Medical Psychology, University Medical Center Hamburg-Eppendorf, Hamburg, Germany

## ABSTRACT

**Background**. The use of the internet to satisfy information needs is widespread among cancer patients. Patients' decisions regarding whether to act upon the information they find strongly depend on the trustworthiness of the information and the medium. Patients who are younger, more highly educated and female are more likely to trust online information. The objectives of this systematic review were to examine the extent to which cancer patients trust in cancer-related online information, internet websites as a source of cancer-related information or the internet as a medium of cancer information.

**Methods**. A systematic review was conducted using five databases (PROSPERO registration number: CRD42017070190). Studies of any kind were included if they measured cancer patients' trust in online health information. Study quality was assessed using the Research Triangle Institute (RTI) item bank. A narrative synthesis was undertaken to examine the included studies.

**Results**. Of the 7,314 citations obtained by the search, seven cross-sectional studies were included in the synthesis. A total of 1,054 patients reported having some or a great deal of trust in online cancer information; 154 patients reported moderately trusting such information; and 833 patients reported having no or little trust in online cancer information, internet websites as a source of cancer-related information or the internet as a medium of cancer-related information. Two of the seven studies reported between group comparisons for the above-stated patient characteristics. The methodological quality of the included studies was diverse.

**Conclusion**. The results of the included studies indicates that approximately half of cancer patients appear to trust cancer-specific online information, internet websites as a source of cancer-related information or the internet as an information medium. However, the small number of included studies, high heterogeneity of participants, methods and outcomes calls for further systematic research. It is important to understand that cancer patients do and will increasingly use trusted cancer information websites to search for information concerning their disease. Therefore, physicians and other health care providers should provide more support and advice to these patients.

Corresponding author
Lukas Lange, lu.lange@uke.de

## INTRODUCTION

Information needs are among the most prevalent unmet supportive care needs of cancer patients throughout their cancer journey (*Harrison et al., 2009*). The majority of cancer patients want to have all available information concerning their illness and treatment (*Davies et al., 2008*; *Jenkins, Fallowfield & Saul, 2001*; *Newnham et al., 2006*). The internet allows cancer patients to fulfill their needs for information regarding their diagnosis, prognosis or likelihood of cure, disease stage, and treatment options or the side effects of the treatment in question (*Castleton et al., 2011*; *Maddock et al., 2011*; *Tariman et al., 2014*). Internet utilization is widespread in advanced economies: 89% of the US population and 80% of the European population reported using the internet at least occasionally (*Poushter, 2016*). Compared with other information sources, the internet has the unique advantage of convenience. Cancer patients can anonymously access the internet anytime from almost anywhere (*Maddock et al., 2011*; *Ziebland et al., 2004*).

The prevalence of cancer patients who used the internet to look for cancer-related information in a Dutch sample, two American samples and a Swedish sample ranged from 60% to 75% (*Castleton et al., 2011*; *Mattsson et al., 2017*; *Mayer et al., 2007*; *Van de Poll-Franse & Van Eenbergen, 2008*), and the prevalence appears to be increasing (*Finney Rutten et al., 2016*).

There are various motivational reasons for cancer patients to search for cancer information on the internet. Patients reported going online because they wanted to develop questions to discuss with their physician, verify information given by their physician, or seek alternative treatments (*Castleton et al., 2011*) and because they felt that the amount of information they received from their physician was insufficient (*Chen & Siu, 2001*).

Information obtained from the internet can impact clinical care in different ways. Cancer-related online information can change patients' choice of treatment, their choice of physician, and their decisions regarding enrollment in a clinical trial (*Castleton et al., 2011*). Most cancer patients do not believe that online information searching negatively affects the doctor-patient relationship (*Newnham et al., 2006*). However, some cancer patients are careful about discussing online information with their physicians. These patients worry that their online searches might have a negative influence on their relationship and might cause physicians to treat them as a problematic patient (*Broom, 2005*; *Chiu, 2011*). Additionally, some oncologists admit to having some difficulty discussing internet-based information with their patients. These oncologists are more likely to report that information obtained from the internet confuses patients (*Helft, Hlubocky & Daugherty, 2003*).

Cancer patients who search the internet for cancer-related information tend to be younger and more highly educated than those who do not (*Castleton et al., 2011*; *Mattsson et al., 2017*; *Shahrokni, Mahmoudzadeh & Lu, 2014*), and they are more likely to have a partner (*Mattsson et al., 2017*). Both age and education are also associated with higher unmet information needs among cancer patients (*Sondergaard et al., 2013*). One study found that female gender was a factor associated with seeking cancer-related information (*Mayer et al., 2007*), while another study did not (*Castleton et al., 2011*). A higher likelihood

of internet use was further associated with better self-reported health among cancer survivors (*Chou et al., 2011*).

The quality of the cancer-related information that can be accessed on the internet is highly variable. Several studies used the DISCERN scale, a validated instrument developed to help consumers evaluate the quality of health-related information on treatment choices for a specific health problem (*Charnock et al., 1999*), to evaluate the quality of websites that provide cancer-related information. The websites were generated by typing cancer-related search terms (i.e., 'cancer', 'cancer therapy', 'breast cancer' or 'colon cancer') in popular search engines, such as Google or Bing and focusing mainly on the first search results (*Borgmann et al., 2016*; *Bruce et al., 2015*; *Hargrave, Hargrave & Bouffet, 2006*; *Liebl et al., 2015*; *Nghiem, Mahmoud & Som, 2016*; *Ni Riordain & McCreary, 2009*; *Wasserman et al., 2014*), as most users will not proceed any further (*Eysenbach & Köhler, 2002*). The evaluated websites' information was often incomplete and did not provide all of the details necessary to allow cancer patients to make well-informed decisions (*Al-Bahrani & Plusa, 2004*; *Borgmann et al., 2016*; *Bruce et al., 2015*; *Hargrave, Hargrave & Bouffet, 2006*; *Liebl et al., 2015*; *Nghiem, Mahmoud & Som, 2016*; *Ni Riordain & McCreary, 2009*; *Wasserman et al., 2014*). However, the results of two studies indicate a difference between the information quality of different website types. The quality of websites from nonprofit organizations or the government was higher than the quality of websites from the medical practices or commercial health information websites (*Liebl et al., 2015*; *Nghiem, Mahmoud & Som, 2016*).

Few studies have investigated which types of websites cancer patients visit to satisfy their information needs. Nonetheless, cancer patients consider health information websites to be a more valuable source of health information than forums or blogs (*Mattsson et al., 2017*). These patients report preferring to obtain reliable information regarding cancer from websites of their oncologist, hospital, or cancer society and are less likely to access websites with a profit interest. However, the same patients mostly accessed websites that were financed and created by pharmaceutic industries (*Van de Poll-Franse & Van Eenbergen, 2008*), which might promote their own interests and can be of lower quality than nonprofit websites (*Liebl et al., 2015*). Additionally, studies have revealed that health seekers do not consistently check the source and date of the health information they found online (*Eysenbach & Köhler, 2002*; *Fox, 2006*).

The varying quality of the cancer-related information available online presents cancer patients with significant challenges in evaluating and selecting reliable online information sources and, more specifically, in assessing the credibility and trustworthiness of these sources. Trust is an important factor associated with the intention to use information found on a website (*Dutton & Shepherd, 2006*; *Lemire et al., 2008*). People who trust in online health information become motivated to participate in various online health-related activities that meet their informational and emotional needs (*Fisher et al., 2008*).

The decision whether to trust a cancer-related online information can be a complex one as information-searching skills, prior experiences with the source and medium of the information, and characteristics of the source may influence the cancer patient's decision. There are different levels (individual, interpersonal, relational and societal) of trust that

have been studied in the literature. The interpersonal level appears to be the appropriate level for determining trust in online information, as information is provided by an author (trustee) and communicated over a certain channel (the internet) to a receiver (trustor) (*Kelton, Fleischmann & Wallace, 2008*). A definition of trust that is often used at this level is that: "The willingness of a party to be vulnerable to the actions of another party based on the expectation that the other will perform a particular action important to the trustor, irrespective of the ability to monitor or control that other party" (*Mayer, Davis & Schoorman, 1995*).

The terms trust and credibility are often used interchangeably. However, credibility can be described as perceived information quality, or the evaluation of information quality by a user (*Fogg & Tseng, 1999*). After evaluating the credibility of information, a reader may decide whether to trust or not trust it. The credibility evaluation process in online environments can be explained by a dual processing model (*Metzger, 2007*) or the 3S-model (where the three "Ss" stand for semantics, surface, and source features of information) (*Lucassen & Schraagen, 2011*). The dual processing model states that the decision whether a heuristic (peripheral) or systematic (central) evaluation is performed is decided by the users' motivation and ability. The users' motivation results from the consequence of receiving inferior, unreliable or inaccurate information online (*Metzger, 2007*). We prefer the 3S-model, because we believe that due to existential concerns and their need for hope, cancer patients are a vulnerable population (*Davey, Butow & Armstrong, 2003*) and therefore would be highly motivated. The 3S-model asserts that the most direct strategy for evaluating credibility is to search for semantic cues (factual accuracy, neutrality or completeness of the information) in the information (*Lucassen & Schraagen, 2011*). However, cancer patients usually search for information that is new to them and thus do not always have the necessary expertise to evaluate the semantics of the information and consequently revert to surface cues (writing style, text length or number of references) (*Lucassen & Schraagen, 2011*). Systematic and heuristic processing are thus both used within a single search process. Additionally, trust in online information is influenced by trust in its source (website), which in turn is influenced by trust in the medium (internet) of this source and a general propensity to trust (*Lucassen et al., 2013*). Users with low trust in the source (website) cannot distinguish between high quality and inferior information (*Lucassen & Schraagen, 2012*). Trust in the internet is largely affected by prior experience with this medium (*Dutton & Shepherd, 2006*). In this study, we are interested in cancer patients' trust in the cancer-related information online, their trust in certain cancer information websites (source) as well as their trust in the internet as a medium of cancer information.

Consumers' trust in health information websites can be influenced by various factors. Two systematic reviews (*Kim, 2016*; *Sbaffi & Rowley, 2017*) assessed these factors, which can be organized into three categories as follows. (1) Individual consumer characteristics: consumers who are younger, are mostly highly educated, are female, have a higher level of agreeableness, have a higher income, reported being in good health condition and have a higher level of health literacy appear to be more trusting of health information websites. (2) Website-related factors: websites that are complete, understandable, unbiased, modern,

useful, and easy to navigate; have a clear and professional layout; are easy to access; are run by medical universities or the federal government; and contain high-quality information are more likely to be trusted. (3) Consumer-to-website interaction-related factors: experience in using the internet and familiarity with the website are likely to influence consumer trust in the health information. Experienced users are more confident in the internet and less concerned over the risks entailed in its use, thereby increasing the likelihood of trusting health-related websites. Additionally, patients prefer health information written by people experiencing similar health issues (*Kim, 2016*; *Sbaffi & Rowley, 2017*).

In summary, it can be stated that cancer patients can only benefit from online cancer information if they can trust the information or the internet as a medium of this information. To date, no systematic review that analyzed cancer patients' trust in online health information has been published. The primary goal of this systematic review is to identify the extent to which cancer patients trust cancer-related online information, internet websites as a source of cancer-related information or the internet as a medium of cancer-related information. As a secondary goal, the review seeks to determine whether trust in cancer-related online information differs across patients of different ages, genders, health statuses, education levels or cancer types.

## METHODS

A systematic literature review was conducted to explore cancer patients' trust in cancer-related online information, internet websites as a source of cancer-related information or in the internet as a medium of cancer information. The protocol for the systematic review was registered in the international prospective register of reviews (PROSPERO) with the registration code CRD42017070190 (File S1). Additionally, the reporting of this review followed the recommendations of the PRISMA statement (*Moher et al., 2009*), an evidence-based minimum set of items for reporting in systematic reviews and meta-analyses (Table S1).

### Information sources and search strategy

We performed an electronic literature search of the electronic databases Medline, CINAHL, Web of Science, PsycINFO and PSYINDEX using prespecified search terms with no restriction on the publication period. All searches were performed on the 4th of January 2017 (last update, 4th of October 2018). Additionally, the reference lists of the included studies were manually searched for potentially relevant studies.

To systematically identify search terms that could address the research questions, the PICO criteria were adapted. PICO criteria can help facilitate the process of finding an answer to a clinical question, by identifying appropriate keywords that can be used to conduct a literature search (*Richardson et al., 1995*; *Van Loveren & Aartman, 2007*). The focus was on the following criteria: (P) population (cancer patients); and (O) outcome (trust/distrust/evaluation of credibility). We included all studies independent of being an intervention study or the presence or absence of a comparison group. Therefore, we did not specify the intervention (I) or comparison (C) in our research question. Additional search terms were selected after an analysis of the Medical Subject Headings (MeSH) and

text words used in key articles, which were identified in prior nonsystematic exploratory literature searches. The included search terms were discussed by the authors and then arranged to create a search string. The search strings were used in each database and accounted for synonyms, plurals, hyphenations and multiple word combinations. All search results were exported into EndNote X7, and all duplicates were removed. The search strategy for MEDLINE is provided in Table S2. The search strategy was appropriately modified for each database to identify eligible studies.

## Eligibility criteria

All studies obtained from the initial search had to fulfill the inclusion criteria of the two selection phases. During the first phase, the corresponding author screened the titles and abstracts of all studies. Consistent with the broad research questions, studies of any type were included if the study title or abstract stated that cancer patients or cancer survivors of any age and with any type of cancer participated in the study and if one of the reported outcomes appeared to be participants' trust, perceptions of credibility or distrust in online cancer-related information, internet websites as a source of cancer-related information or in the internet as a medium of cancer information.

During the second phase, two researchers independently assessed the full texts of the remaining potentially relevant articles. The eligibility criteria used in the full text screening addressed two aspects: study characteristics and report characteristics (*Liberati et al., 2009*). The inclusion criteria for study characteristics were as follows: (1) the full text was available; (2) any study type was included if it included some form of quantitative data; (3) at least some of the participants were cancer patients or cancer survivors; (4) the participants were 18 years or older; (5) the measured constructs were trust, perceived credibility or distrust; and (6) the study measured participants' trust in online cancer-related information, internet websites as a source of cancer-related information or the internet as a medium of cancer-related information. The reporting of the study had to meet one inclusion criterion: (1) the study was included in the review if it was reported in English or German. Disagreements between the researchers regarding the eligibility of studies were resolved via discussion. The reasons for exclusion and the number of studies excluded for each reason can be found in Table S3.

## Data extraction and quality assessment

Data extraction was performed by the corresponding author (LL) and cross-checked by another member (MLP) of the research group. The following data were extracted from the included studies: (1) study characteristics, including the author name, year of publication, title of publication, place of data collection, study design, and sample size; (2) characteristics of the study participants, such as age, gender, cancer type, education status, health status; (3) outcome characteristics, such as questionnaire or items used to measure trust; and (4) the measured outcome of trust, perceived credibility or distrust (i.e., as the mean or distribution). Additionally, six corresponding authors of the studies in question were contacted for further information (e.g., questionnaires, data sets); four of them responded. Most provided additional information concerning their publications.

Three authors provided additional descriptive information, and the fourth author shared the complete data file of the study results.

The methodological quality assessment of the included papers was independently performed by two researchers in the study group and based on the RTI item bank (*Viswanathan et al., 2013*). The RTI item bank provided the researchers with a set of items to evaluate the conduct of observational studies included in systematic reviews and to detect possible risks of biases of the included studies (*Viswanathan & Berkman, 2012*; *Viswanathan et al., 2013*). Studies were not excluded from the review or any subsequent analyses on the basis of the risk of bias. In accordance with the developers' instructions, the instrument was adapted to fit the designs of the included observational studies. Seven questions (questions 1, 2, 3, 6, 9, 11, 13) were used to assess selection bias, detection bias, confounding, selective outcome reporting and overall bias in the included studies. The reasons for not integrating the additional questions of the RTI item bank into the quality assessment can be found in Table S4. Disagreements between the two reviewers regarding the assessed quality of the studies were resolved via discussion.

## Data analysis and description

A narrative synthesis was undertaken to examine the included studies (*Dixon-Woods et al., 2005*). Furthermore, the characteristics and results of the included studies were summarized descriptively.

No planned meta-analysis was conducted to answer the secondary study goal (does trust in online cancer-related information, internet websites as a source of cancer-related information or in the internet as a source medium of cancer information differ across patients with different ages, genders, education levels, health status or cancer types) because only two of the included studies reported between-group comparisons for these patient characteristics. However, to illustrate the between-group comparisons of individual studies, we calculated the mean differences (MDs) by subtracting the mean score of one group of participants (i.e., female patients), which was expected to score higher, from the mean score of the second group of participants (i.e., male patients). For clarity, all the mean trust scores reported in the included studies were transformed into a 5-point scale (range, 1–5).

## RESULTS

### Study selection

The search of the databases (Medline, CINHAL, Web of Science, PsychINFO and PSYINDEX) resulted in 7,314 citations (Fig. 1). All citations are available at Figshare (https://doi.org/10.6084/m9.figshare.7701014.v1). After the removal of duplicate articles, 6132 titles and abstracts were scanned for eligibility. Of these, 54 studies fulfilled the eligibility criteria of the first selection phase. Four additional studies were added after the reference lists of the 54 potentially relevant studies were manually searched. The full text of two articles could not be retrieved despite contacting the authors. Of the remaining 51 articles, seven (*Crutzen et al., 2014*; *Losken et al., 2005*; *Lussiez et al., 2017*;
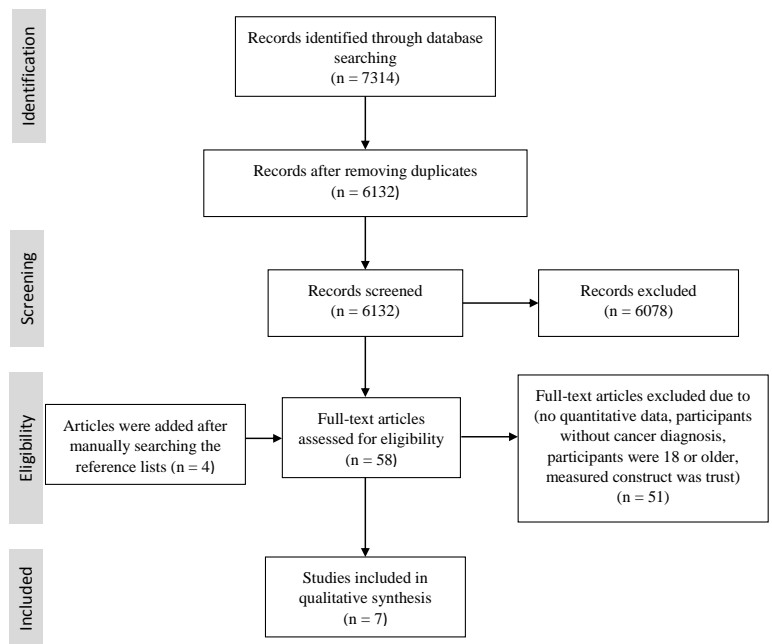

**Figure 1   Flowchart of the literature search.**

*Mayer et al., 2007*; *Pereira et al., 2000*; *Roach et al., 2009*; *Shea-Budgell et al., 2014*) met the eligibility criteria of the second phase and were therefore included in the review.

## Study characteristics

The characteristics of all the articles are described in Table 1. All studies were published in English in peer reviewed journals between 2000 and 2014. Six of the seven studies were performed in North America (USA and Canada), and one (*Crutzen et al., 2014*) was conducted in Europe (Netherlands). Four studies used on-site recruitment of participants visiting outpatient facilities for treatment or checkups (*Losken et al., 2005*; *Lussiez et al., 2017*; *Pereira et al., 2000*; *Shea-Budgell et al., 2014*). Two studies (*Mayer et al., 2007*; *Roach et al., 2009*) called patients at their homes, and one study (*Crutzen et al., 2014*) recruited participants online. All seven studies were cross-sectional. Five of the studies performed out to answer broad research questions, while two studies (*Lussiez et al., 2017*; *Roach et al., 2009*) formulated hypotheses. Three of the studies formulated the intention to describe the general internet use of patients with any type of cancer (*Losken et al., 2005*; *Pereira et al., 2000*; *Shea-Budgell et al., 2014*). One study intended to provide insights into user perceptions related to loyalty towards a specific Dutch cancer information website (*Crutzen et al., 2014*). The website offered tailored, hospital-specific information on oncological care and detailed information about health care professionals, and it was constantly reviewed and updated by professionals in oncology care. Another study aimed to describe differences between cancer survivors who do and do not seek cancer information off- or online (*Mayer et al., 2007*). The authors of the sixth and seventh study hypothesized that cancer patients are more likely to search for cancer information (*Lussiez et al., 2017*; *Roach et al., 2009*), higher

income and education levels correlate with increased internet use for health information (*Lussiez et al., 2017*) and cancer patients are more likely to have an increased knowledge of cancer-related information sources than healthy individuals (*Roach et al., 2009*).

All but one study used one item scored on a 3- to 5-point scale to measure the participants' trust in information found on the internet (*Losken et al., 2005*; *Lussiez et al., 2017*; *Pereira et al., 2000*) or the internet as a medium of health information (*Mayer et al., 2007*; *Roach et al., 2009*; *Shea-Budgell et al., 2014*). The seventh study used 3 items scored on a 7-point Likert scale to measure patients' trust in one specific cancer information website (*Crutzen et al., 2014*). Convenience and purposive sampling methods were applied in five studies, while two studies (*Mayer et al., 2007*; *Roach et al., 2009*) applied stratified sampling with oversampling of minority groups. The sample sizes of cancer patients in the included studies ranged from 63 to 719 (median, 157).

## Synthesis of the results

The study results were grouped into three themes based on the study goals: (1) characteristics of the included study samples; (2) cancer patients' trust in cancer- or health-specific online information, in internet websites as a source of cancer-related information or in the internet as a medium of cancer information; and (3) between-group comparison of patient characteristics (age, gender, education and cancer type). A full overview of the results is displayed in Table 2.

## Sample characteristics of the included studies

In the five studies that reported the participants' age, the mean age was 56 years (range, 47–58). In the five studies that reported the participants' education levels, the largest group in each study was highly educated. Two studies did not report their participants' education levels (*Lussiez et al., 2017*; *Roach et al., 2009*).

Three studies (*Mayer et al., 2007*; *Roach et al., 2009*; *Shea-Budgell et al., 2014*) included cancer patients with various types of cancer, while three studies (*Losken et al., 2005*; *Lussiez et al., 2017*; *Pereira et al., 2000*) focused on particular types. Two studies (*Losken et al., 2005*; *Pereira et al., 2000*) focused on breast cancer patients, while one (*Lussiez et al., 2017*) focused on lung or esophageal cancer. One study did not report the cancer types of the included participants (*Crutzen et al., 2014*). Six of the seven studies that reported the participants' gender included more female than male participants (63% females among the 1,347 participants).

## Cancer patients' trust in online cancer information

Approximately half of the 2,041 cancer patients who participated in all of the included studies combined appeared to trust online cancer information ($\bar{x} = 3.1$, SD $= 1.4$). A total of 1,054 patients reported having some or a lot of trust in cancer information obtained online, while 154 patients reported moderately trusting or being undecided about whether they should trust online health information, and 833 patients reported having no or a little trust in cancer-related information found on the internet or in the internet as a medium of cancer-related information.

Lange et al. (2019), *PeerJ*, DOI 10.7717/peerj.7634

**Table 1  Study characteristics of the included cross sectional studies.**

| Study | Title | Year | Country | Study design | Recruitment strategy | Instruments to measure trust | Sampling method |
|---|---|---|---|---|---|---|---|
| *Crutzen et al. (2014)* | E-loyalty towards a cancer information website: applying a theoretical framework | 2014 | NL | CS | Online invitations to visit the website & asking visitors of the website to evaluate it | Three items scored on a 7-point Likert items ('strongly disagree' to 'strongly agree') to measure trust in one specific website | Nonprobability sampling: convenience sampling & purposive sampling |
| *Losken et al. (2005)* | Infonomics and breast reconstruction—Are patients using the internet? | 2005 | USA | CS | On-site recruitment (first post-operative visit) | One item scored on a 3-point item format ('disagree' to 'agree'): "Did you trust the information" found in the internet | Nonprobability sampling: purposive sampling |
| *Lussiez et al. (2017)* | Internet usage trends in thoracic surgery patients and their caregivers | 2015 | USA | CS | On-site recruitment (outpatient clinic visit) | One item scored on a 3-point item format ('not trustful' to 'very trustful') to measure the "level of trust subjects placed in the information they found" online | Nonprobability sampling: purposive sampling |
| *Mayer et al. (2007)* | Cancer survivors information seeking behaviors: | 2007 | USA | CS | Called via telephone | One item scored on a 4-point item format ('not at all' to 'a lot') to measure "how much they trusted different sources of information" (i.e., internet) | Stratified sampling with oversampling of minority groups |

Lange et al. (2019), *PeerJ*, DOI 10.7717/peerj.7634

**Table 1** (*continued*)

| Study | Title | Year | Country | Study design | Recruitment strategy | Instruments to measure trust | Sampling method |
|---|---|---|---|---|---|---|---|
| *Pereira et al. (2000)* | Internet usage among women with breast cancer: an exploratory study | 2000 | Canada | CS | On-site recruitment (outpatient clinic visit) | One item scored on a 5-point Likert item ('completely disagree' to 'completely agree'): "I trust the medical information I found on the Internet." | Nonprobability sampling: purposive sampling |
| *Roach et al. (2009)* | Differences in cancer information-seeking behavior, preferences, and awareness between cancer survivors and healthy controls | 2009 | USA | CS | Called via telephone | One item scored on a 4-point item format ('not at all' to 'a lot') to "indicate how much they would trust cancer information obtained from the following sources" (i.e., internet). | Stratified sampling with oversampling of minority groups |
| *Shea-Budgell et al. (2014)* | Information needs and sources of information for patients during cancer follow-up | 2014 | Canada | CS | On-site recruitment (outpatient clinic visit) | One item scored on a 4-point item format ('not at all' to 'a lot' = 4) to measure "the level of trust in information sources" | Nonprobability sampling: purposive sampling |

**Notes.**

CS, Cross-sectional; NL, Netherlands.

[a]Number of particpants included in the study.

**Table 2  Outcomes of the included studies.**

| Study | Trust[a] (m, SD) | Sample characteristics | | | | | Differences in trust between groups (MD, [95% CI], Cohen's *d* effect size) | | | |
|---|---|---|---|---|---|---|---|---|---|---|
| | | N[b] | Age[c] (m/SD) | Education | Cancer type | Gender (% female) | Female vs. male patients | Highly vs. low educated | Young vs. old patients | Cancer types |
| *Crutzen et al. (2014)* | 3.8 (1.0) | 45 | 53 (12) | 46% high education; 40% intermediate education; 14% low education | NR | 76 | 0.13, [−0.45, 0.71], *d* = 0.14 | −0.58, [−1.64, 0.48], *d* = 0.58 | NR | NR |
| *Losken et al. (2005)* | 4.3 (1.0) | 72 | 50 (30–70) | 22% graduate; 43% college; 30% some college, 14% high school | Breast cancer patients | 100 | NR | NR | NR | NR |
| *Lussiez et al. (2017)* | 3.9 (1.1) | 192 | 54 (14–86) | NR | Lung or esophageal cancer | 55 | NR | High educated participants reported higher trust scores | NR | NR |
| *Mayer et al. (2007)* | 2.9 (1.6) | 597 | 58 | 44% more than high school; 38% high school; 18% less than high school | 18 different cancer types[d] | 65 | NR | NR | NR | NR |
| *Pereira et al. (2000)* | 3.5 (0.6) | 32 | 47 (9) | 66% college or university; 34% junior or senior high school | Breast cancer patients | 100 | NR | NR | NR | NR |
| *Roach et al. (2009)* | 3.0 (1.6) | 692 | NR | NR | Different cancer types[e] | NR | NR | NR | NR | NR |
| *Shea-Budgell et al. (2014)* | 3.1 (1.2) | 411 | NR | 33% high school or less; 21% post–high school; 33% college or university; 10% postgraduate; 4% not specified | Seven different types/group of cancer types reported | 53 | NR | NR | NR | No significant associations between cancer site and level of trust |

**Notes.**

m, mean; SD, standard deviation; MD, mean difference; NR, not reported.

[a]Average trust in online information reported on a 5-point item format.

[b]The number participating of cancer patients.

[c]Age reported in years.

[d]Breast and cervical were most frequent reported types of cancer.

[e]The most common initial diagnoses were gynecologic, non-melanoma skin, and breast cancers.

Regarding the individual results of each study, two points are apparent. In all studies but one (*Roach et al., 2009*), the majority of the participants reported having some or a lot of trust in online cancer information. Additionally, 55% of breast cancer patients in outpatient treatment stated that they were undecided about whether to trust online information (*Pereira et al., 2000*).

Three of the included studies indicated that the internet appears to be the second-most-trusted medium of cancer information after health care professionals (*Mayer et al., 2007*; *Roach et al., 2009*; *Shea-Budgell et al., 2014*). Patients were more likely to trust online information than information from newspapers, magazines, the radio, family or friends or the television.

## Between-group comparisons of patient characteristics

Two of the seven studies reported between-group comparisons of characteristics that might influence patients' trust in online health information. One study found that participants were more likely to be very trustful of information found online if they had a higher education level (*Lussiez et al., 2017*). The authors of the second study stated that no significant associations were detected between cancer site and level of trust (*Shea-Budgell et al., 2014*). The conclusions of both studies could not be verified or replicated because the necessary descriptive data were not reported.

Furthermore, the corresponding author of one study (*Crutzen et al., 2014*) provided his entire data set, which made it possible to calculate two additional results. The MD of the reported trust between patients with high and low education levels was $-0.58$ (95% confidence interval (CI): $-1.64$, $0.48$) with a medium effect size; between female and male patients, the MD was $0.13$ (95% CI [$-0.45$, $0.71$]), with a small effect size (*Crutzen et al., 2014*).

Additionally, two studies reported between-groups comparisons of patients' trust as related to patient characteristics that were not part of the secondary research question. The MDs in the reported trust between cancer patients and healthy control groups were $-0.03$ (95% CI [$-0.33$, $0.27$]) (*Roach et al., 2009*) and $-0.07$ (95% CI [$-0.21$, $0.06$]) (*Crutzen et al., 2014*), with a small effect. The MD between patients who had personally sought cancer information and those who had not was $1.14$ (95% CI [$0.88$, $1.40$]), with a large effect size (*Mayer et al., 2007*).

## Quality assessment

The detailed methodological quality ratings of the included studies are displayed in Table 3. An estimation of the selection bias is covered by questions one to three. The inclusion criteria did not vary across individuals in three of the included studies (question 1). However, four studies failed to report the inclusion criteria (*Crutzen et al., 2014*; *Lussiez et al., 2017*; *Mayer et al., 2007*; *Roach et al., 2009*). None of the studies used different recruitment measures across individuals (question 2). The selection of an appropriate comparison group was not relevant to the included studies as none of them included a comparison group (question 3).

Question six gives an indication of the detection bias of the included studies. Three studies (*Crutzen et al., 2014*; *Mayer et al., 2007*; *Roach et al., 2009*) used valid and reliable

**Table 3  Risk of bias appraisal using the RTI item bank.**

| Study | Selection bias/confounding | | | Detection bias | Selective outcome reporting | Confounding | Overall Assessment |
|---|---|---|---|---|---|---|---|
| | Q1 | Q2 | Q3 | Q6 | Q9 | Q11 | Q13 |
| *Crutzen et al. (2014)* | NR | No | NA | Yes | No | Yes | No |
| *Losken et al. (2005)* | No | No | NA | No | No | Yes | Yes |
| *Lussiez et al. (2017)* | NR | No | NA | No | No | Yes | No |
| *Mayer et al. (2007)* | NR | No | NA | Yes | No | Yes | No |
| *Pereira et al. (2000)* | No | No | NA | No | No | Yes | No |
| *Roach et al. (2009)* | NR | No | NA | Yes | No | Yes | No |
| *Shea-Budgell et al. (2014)* | No | No | NA | NR | No | Yes | Partially |

Notes.
Q1, Do the inclusion criteria vary across the participants of the study?; Q2, Does the strategy for recruiting participants into the study differ?; Q3, Is the selection of the comparison group inappropriate?; Q6, Were valid and reliable measures used?; Q9, Are any important primary outcomes missing from the results; Q11, Are results believable taking study limitations into consideration?; Q13, Were the important confounding variables taken into account in the design?; NR, not reported; NA, not applicable.

instruments, while the remaining four studies used self-developed questionnaires that were not psychometrically validated. Question nine asks whether the researchers were selective in their outcome reporting. None of the seven studies failed to report the results of any of the important primary outcomes. Confounding was accounted for in all studies by believably taking study limitations into consideration (question 11). Question thirteen asks whether important confounding variables were taken into account in the design and/ or the analysis. Five of the studies did take important confounding variables into account. One study reported the descriptive results of possible confounders such as age or education, but did not investigate whether any of the confounders had an influence on the measured outcome (*Losken et al., 2005*). Another study did not state whether the types of information the patients sought varies by any socioeconomic factors (*Shea-Budgell et al., 2014*).

## DISCUSSION

This review includes seven studies that describe cancer patients' trust in cancer-related online information. Overall, approximately half of the cancer patients in the included studies reported that they trusted cancer-related online information, internet websites as a source of cancer-related information or the internet as a medium of cancer-related information.

There appears to be differences in the results of the seven included studies. The lowest trust scores were reported in the three studies with the largest samples of cancer patients with various types of cancer (*Mayer et al., 2007*; *Roach et al., 2009*; *Shea-Budgell et al., 2014*). In these studies, the participants were asked to appraise their trust in the internet as a medium of cancer-related information. In three of the other four studies (*Losken et al., 2005*; *Lussiez et al., 2017*; *Pereira et al., 2000*), the participants were asked how much they trusted the cancer- or health-specific information they found online. The seventh study measured patients' trust in one specific cancer information website (*Crutzen et al., 2014*). Possible explanations for the differences in trust reported by the groups of participants

in these studies might be the formulation of the items that measured patients' trust or the gender of the study participants. When patients are asked to report how much they trust the information they find online, they might be more likely to interpret the question as a rating of their information literacy (*Shenton, 2009*) or eHealth literacy (*Norman & Skinner, 2006*), which are defined as the skills needed to find, retrieve and analyze information in general (information literacy) or health information online (eHealth literacy) and use it appropriately. The potential bias that occurs when patients rate their own information literacy or eHealth literacy is that people are likely to overestimate their own abilities (*Merritt, Smith & Di Renzo, 2005*; *Mohmood, 2016*; *Van der Vaart et al., 2011*). Furthermore, two of the studies with higher reported trust scores (*Losken et al., 2005*; *Pereira et al., 2000*) mainly focused on breast cancer patients. These studies reported mean trust levels of 4.3 and 3.5, which is in line with prior research that stated that female consumers are more likely to trust online information (*Kim, 2016*; *Sbaffi & Rowley, 2017*). Nevertheless, a between-group comparison within the studies showed no difference between female and male patients.

The internet appears to be the second-most-trusted cancer information medium behind health care professionals (*Mayer et al., 2007*; *Roach et al., 2009*; *Shea-Budgell et al., 2014*). Patients are more likely to trust online information than information from newspapers, magazines, the radio, family or friends or the television. A possible explanation might be that the internet offers information that has greater relevance to the consumer and therefore is considered more trustworthy (*Song & Zahedi, 2007*). Magazines, radio and television present information that might target cancer patients or patients with a certain cancer type, whereas the internet allows cancer patients to search for information that is relevant to their individual situation or question.

The number of breast cancer patients in one of the included studies (*Pereira et al., 2000*) that reported that they were undecided regarding whether they should trust the medical information they found online was higher than expected. A possible explanation might be the varying quality of online information, which makes it difficult for cancer patients to make well-informed medical decisions (*Al-Bahrani & Plusa, 2004*; *Borgmann et al., 2016*; *Bruce et al., 2015*; *Hargrave, Hargrave & Bouffet, 2006*; *Liebl et al., 2015*; *Nghiem, Mahmoud & Som, 2016*; *Ni Riordain & McCreary, 2009*; *Wasserman et al., 2014*). Physicians and oncologists should adopt an intermediary role when they discuss internet information with their patients. They should be able recommend reliable online information sources to their patients and help them understand and discuss the information found there (*Halwas, Griebel & Huebner, 2017*). Patients who search for information should not be viewed as a threat (*Helft, Hlubocky & Daugherty, 2003*) but as an opportunity to increase communication and shared decision-making ability (*Kehl et al., 2015*).

It was not possible to answer the secondary research question: Does trust in online health information differ between cancer patients with different ages, genders, health status or education levels? Most of the included studies did not examine how patients' characteristics influence the amount of reported trust in online health information as measuring trust was not their main research goal. The results of the included studies that did examine within-group differences did not always confirm the results of recent systematic reviews

that identified consumer characteristics that might influence the reported trust in health information websites (*Kim, 2016*; *Sbaffi & Rowley, 2017*): In one study, as expected, female patients reported higher levels of trust (*Crutzen et al., 2014*). The effect size for the MD between females and males was quite small. The results of the comparisons between patients with higher and lower education levels within the studies were mixed. One study (*Lussiez et al., 2017*) reported the expected difference, while the other study (*Crutzen et al., 2014*) stated that patients with lower education levels were more likely to trust online information. There appears to be no explanation for the higher mean trust scores of the low-education patient groups compared with the high-education patient groups. However, the study sample of cancer patients with low education levels only included five individuals (*Crutzen et al., 2014*) and therefore may not represent the entire population of cancer patients with low education levels. A within-study comparison indicated that cancer type appeared to have no influence on the reported trust in online cancer information (*Shea-Budgell et al., 2014*). The absence of a difference among different cancer types did not contradict any study findings. Cancer type was added to the list of possible confounding variables because a difference in reported trust was expected due to differences in attitudes towards eHealth among patients with different types of cancer (*Jansen et al., 2015*). The MD in reported trust between patients who had personally sought cancer information and those who had not had a large effect size. This effect may be explained by differences in research findings. Patients who are searching for cancer information are more likely to have completed a higher level of education (*Ramanadhan & Viswanath, 2006*), which is again associated with a higher tendency to trust online information (*Kim, 2016*; *Sbaffi & Rowley, 2017*). Additionally, experienced users are more confident in the internet and less concerned over the risks entailed in its use, both of which have a positive influence on trust (*Kim, 2016*).

In terms of the methodological quality of the included studies, it should be noted that the assumptions of this review are exclusively based on the results of cross-sectional studies, which are likely to have different biases (*Viswanathan & Berkman, 2012*). Overall, the assessment of the included studies with the RTI item bank indicated that estimations of selective outcome reporting, confounding and overall assessment gave a positive impression of the methodological quality of the included studies. However, four studies failed to report whether the inclusion criteria varied across participants. Due to the lack of information on inclusion criteria, a selection bias cannot be ruled out which may limit the generalizability of the study results (*Hernán, Hernández-Díaz & Robins, 2004*). However, not reporting the inclusion criteria only allow conclusions to be drawn about the reporting quality of the included studies, but not about the quality of the studies (*Margulis et al., 2014*). Additionally, only three of the studies used valid and reliable instruments, which is an indicator for detection bias (*Viswanathan et al., 2013*). Additional research needs to be conducted using a validated instrument to measure patients' trust in cancer-related information websites.

Six of the studies only used one item to measure the primary outcome of trust, creating a possible source of bias. Latent variables are usually complex and not easily measured. The use of multiple items helps to average out errors that are inherent in single items and therefore have higher reliability and criterion validity than a single item (*Sarstedt*

*& Wilczynski, 2009*). Single items have practical advantages, such as parsimony and ease of administration (*Bergkvist & Rossiter, 2009*), and they usually promote higher response rates (*Bergkvist & Rossiter, 2007*). However, they only perform as well as multi-item scales under very specific conditions (*Diamantopoulos et al., 2012*). Under these conditions, the construct should be unidimensional and unambiguous to the respondent (*Wanous, Reichers & Hudy, 1997*). Examples of these types of constructs are job satisfaction (*Wanous, Reichers & Hudy, 1997*) and attitude towards advertisement and brand in marketing (*Bergkvist & Rossiter, 2007*; *Bergkvist & Rossiter, 2009*). Because assessing trust in information always will contain a certain degree of heuristics (*Lucassen et al., 2013*), it appears as if one item might provide a sufficient indication of whether cancer patient trust online information or the internet as an information source. Nevertheless, there appears to be a need for validated questionnaires that measure consumers' trust in online health information.

## Strengths and limitations

This systematic review has some limitations. Its main limitation is the lack of knowledge of the webpages that formed the basis of the study participants' trust assessments. As stated in the introduction, the quality of the cancer-related information that can be accessed on the internet is highly variable (*Al-Bahrani & Plusa, 2004*; *Borgmann et al., 2016*; *Bruce et al., 2015*; *Hargrave, Hargrave & Bouffet, 2006*; *Liebl et al., 2015*; *Nghiem, Mahmoud & Som, 2016*; *Ni Riordain & McCreary, 2009*; *Wasserman et al., 2014*). When participants had a negative prior experience with cancer websites, they were more likely to report low levels of trust (*Kim, 2016*). Furthermore, only studies written in German and English were included in the study. Therefore, studies in other languages that examine cancer patients' trust in cancer-related online information might be missing from this systematic review. Another limitation could be that six of the seven included studies were conducted in North America, and only one was conducted in Europe, although there appears to be only small differences in the reported internet utilization of these populations (*Poushter, 2016*). We can therefore only draw conclusions about North American cancer patients. No gray literature was included in the review. Therefore, we missed the opportunity to minimize the effects of publication bias and to represent the entire evidence base as studies that show statistically significant, positive results have a better chance of being published (*Blackhall & Ker, 2007*; *Hopewell et al., 2007*).

In addition to the limitations, this review also has distinct strengths. First, because the authors aimed to provide a broad picture of cancer patients' trust in online health information, the study used a systematic search strategy of five electronic databases, which resulted in a heterogeneous sample of studies and did not exclude studies due to their design or quality. Additionally, the search strategy used appeared to be successful as only four additional studies could be identified through manual searches of the reference lists of studies that fulfilled the eligibility criteria of the first selection phase, and none of these four studies was included in the review. A further strength is the methodological quality assessment of the included papers, which was independently performed by two researchers according to the reporting guidelines of the PRISMA statement (*Moher et al., 2009*). Finally, this systematic review was registered in PROSPERO to achieve transparency.

## CONCLUSIONS

This systematic review included seven cross-sectional studies out of 7,314 citations obtained from a search. The results of the included studies indicate that approximately half of cancer patients appear to trust information found on the internet, internet websites as a source of cancer-related information or trust the internet itself as a medium of cancer information. However, the small number of included studies, high heterogeneity of participants, methods and outcomes, and the diverse quality of the included studies call for further systematic research.

Further research on cancer patients needs to be conducted using a validated instrument to measure patients' perceived trust and the credibility of health information websites, especially for groups of patients such as older adults and those with a low socioeconomic status, who appear to have lower online information searching skills and tend to be less likely to trust cancer information found online.

Furthermore, it is important to understand that cancer patients' decision making is influenced by online information and that even if the physician remains the most trusted medium of advice, patients do and will increasingly use cancer-related websites to search information concerning their disease and its treatment. Therefore, physicians, nurses and other health care providers should provide more support and advice to patients seeking health information. Additionally, patients should be encouraged to ask their doctors questions and to discuss the results of their online information searches with them to ensure that false information is not included.

## ACKNOWLEDGEMENTS

The authors of this systematic review wish to acknowledge the authors of the included studies for their cooperation.

### Funding
The authors received no funding for this work.

### Competing Interests
The authors declare there are no competing interests.

### Author Contributions
- Lukas Lange conceived and designed the experiments, performed the experiments, analyzed the data, contributed reagents/materials/analysis tools, prepared figures and/or tables, authored or reviewed drafts of the paper, approved the final draft.
- Mona Leandra Peikert performed the experiments, analyzed the data, contributed reagents/materials/analysis tools, authored or reviewed drafts of the paper, approved the final draft.
- Christiane Bleich and Holger Schulz conceived and designed the experiments, authored or reviewed drafts of the paper, approved the final draft.

## Data Availability

The raw data are available at Figshare: Lange, Lukas (2019): RAW Data Systematic Review.xlsx. figshare. Dataset. https://doi.org/10.6084/m9.figshare.7701014.v1.

## Supplemental Information

Supplemental information for this article can be found online at http://dx.doi.org/10.7717/peerj.7634#supplemental-information.

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
