# Peer review of "The extent to which cancer patients trust in cancer-related online information: a systematic review"

_PeerJ, doi:10.7717/peerj.7634_

## Round 0.1 · original submission · Major Revisions

The comments by the referees are not converging (R1 recommended Rejection, and R2 Minor revisions), and so it was not easy to make a final decision. However I believe that there should be a chance to the authors to critically revise the manuscript following, with particular care, referee's 1 comments.

Reviewer 1 ·

Basic reporting

• This manuscript should be reviewed by a native English speaker to review multiple errors (grammar and tenses). The authors should also proof-read the manuscript to correct errors e.g. Line 39: “therefore, physicians and other health care providers have to should…”, Line 370: “…between females and males was quiet small”

• The introduction supports the rationale for the study. However, in line 100, the authors state that credibility is only one trust component but do not elaborate on their stance (definition or description) on trust

• The results are relevant to the hypothesis. The authors rightly point out that the available evidence does not support conclusions on the second part of their hypothesis.

Experimental design

• The research meets the standards for conducting systematic reviews. However, the authors’ method for rating the quality of studies is questionable. The RTI item bank used requires studies to clearly state their inclusion criteria. However, as stated in Line 267, four of the included studies did not provide details on their inclusion criteria. The authors should clarify what they hoped to achieve by using the RTI criteria to assess the studies. They should also provide reasoning for choosing some of the criteria and not others.

• In line 152 the authors state 'participants' trust' and later 'active trust', how do they distinguish between the two?

• In line 156, the full text screening was divided into 'study characteristics' and 'report characteristics'. Yet, the authors do not specify which criteria falls into which group in the subsequent text.

• In line 135, the authors should define what PICOS stands for and their reasoning for choosing certain aspects of the acronym and not others

• There is some redundancy between the information in 'Eligibility criteria' (line 146) and 'data extraction and quality assessment (line 166). E.g. both line 154 and 179 states that two researchers reviewed the studies

• in line 111, the sentence isn't clear and may be misconstrued

Validity of the findings

• The hypothesis of this study is relevant given that it approaches the topic from a systematic review perspective. However, since the quality of the studies included in the review is somewhat questionable, it dilutes the value of conducting a systematic review.

• The conclusion rightly stated that there is little evidence to answer the second question they sought to answer. However, going by Table 2, it appears the other conclusions may be overstated. Rather than attempting to make broad conclusions, the authors should be more subtle about the generalisability of their findings

Reviewer 2 ·

Basic reporting

Good.

Experimental design

Good.

Validity of the findings

Good.

Additional comments

While the Introduction is well-written and covers important research pertaining to cancer information seeking, it does not provide an understanding of what sites cancer patients are most likely to access, e.g. official institutions, cancer charities, online forums for cancer patients, blogs, etc. This is important because we need to have a picture of what information sources cancer patients access online, and how these may differ in their trustworthiness. For example, lines 83-85, page 4, state that “websites’ information was often incomplete”, but there is no detail on what kind of websites these were. If such information is not included in the original studies, the authors of the present review need to say so and add some reflection on the shortcomings of past research.
The Introduction could also benefit from a distinction between trust in online cancer information vs. trust in the Internet, as the former refers to content and the latter, to a channel of information diffusion. Clarifying this aspect would help refine the goal of the present review in lines 116-118.
Another aspect which is underdeveloped is the definition of trust for the purpose of the present review (lines 99-100). The authors need to elaborate on the process of which trust is the outcome, i.e. is this process underpinned by heuristics or considered judgements, or both? And how might we expect heuristics or deliberate analysis to occur in relation to different websites? It may also need to be acknowledged that trust can be subjective and thus fulfil a psychological function (coping with threat). Cancer patients may be in a vulnerable state therefore their willingness to place trust in certain online information sources may be linked to their psychological need to reduce their vulnerability.
Line 221 – in Lussiez et al.’s study, the participants were thoracic surgery patients, so they are not patients with any type of cancer as the authors state.
In the Results, the section on cancer patients’ trust in online cancer information should have included information on how trust was measured in the 7 reviewed studies, i.e. the wording of the questions/scales used.

External reviews were received for this submission. These reviews were used by the Editor when they made their decision, and can be downloaded below.

---

## Round 0.2 · accepted · Accept

The paper by Dr Lange et al, has been strongly recommended for publication by the referee.

Reviewer 2 ·

Basic reporting

I commend the authors for improving this manuscript which I strongly recommend for publication. The manuscript reads much better now and makes a significant contribution to the field, as there is currently a lack of studies which have examined trust in online health information among cancer patients.

Experimental design

I am not familiar with RTI item bank and therefore cannot comment on it. However, I am convinced that the systematic review and appraisal of the studies included were conducted thoroughly.
It is good that the authors deleted the concept of 'active trust' from their manuscript as this term has very little scientific usage (I have never encountered 'active trust' in my own research on trust in online health information).
The authors provide a very clear distinction between trust and credibility and clarify the processes through which cancer patients can evaluate the credibility of online sources (heuristically or systematically) and place trust in them.

Validity of the findings

I think the findings are valid and the conclusions are pertinent. It would have been good if the authors could have included qualitative studies exploring why cancer patients trust or do not trust online health / cancer information, but this systematic review is nonetheless good. The number of studies addressing trust in online health information among cancer patients is relatively small so the authors did well to find any and to synthesise their findings. The authors highlight some of the antecedents of trust in online health information and also highlight the fact that the way researchers measure trust in online health information needs to be improved and made more transparent when reporting research findings. Last but not least, the authors include very good reflection on the strengths and limitations of their systematic review.

Additional comments

In the manuscript the authors provide much needed new insights into an area which is currently under-researched. I will follow with interest the authors' next research and publications and wish them good luck in their new research initiatives.

External reviews were received for this submission. These reviews were used by the Editor when they made their decision, and can be downloaded below.